# Mining and quantitative evaluation of COVID-19 policy tools in China

**Jianzhao Liu☯, Na Li☯, Luming Cheng📄*

School of Economics and Management, Tianjin Chengjian University, Tianjin, China

☯ These authors contributed equally to this work.
* chengluming@yeah.net

**Data Availability Statement:** All relevant data are within the paper and its Supporting Information files.

**Funding:** The National Social Science Fund of China, 18BJL067, A/Prof. Jianzhao Liu.

## Abstract

Policy quantitative analysis can effectively evaluate the government's response to COVID-19 emergency management effect, and provide reference for the government to formulate follow-up policies. The content mining method is used to explore the 301 COVID-19 policies issued by the Central government of China since the outbreak of the epidemic in a multi-dimensional manner and comprehensively analyze the characteristics of epidemic prevention policies. Then, based on policy evaluation theory and data fusion theory, a COVID-19 policy evaluation model based on PMC-AE is established to evaluate quantitatively eight representative COVID-19 policy texts. The results show that: Firstly, China's COVID-19 policies are mainly aimed at providing economic support to enterprises and individuals affected by the epidemic, issued by 49 departments, and include 32.7 percent supply-level and 28.5 percent demand-level, and 25.8 percent environment-level. In addition, strategy-level policies accounted for at least 13 percent. Secondly, according to the principle of openness, authority, relevance and normative principle, eight COVID-19 policies are evaluated by PMC-AE model. Four policies are level I policies, three policies are level II policies and one policy is level III policies. The reason for its low score is mainly affected by four indexes: policy evaluation, incentive measures, policy emphasis and policy receptor. To sum up, China has taken both non-structural and structural measures to prevent and control the epidemic. The introduction of specific epidemic prevention and control policy has realized complex intervention in the whole process of epidemic prevention and control.

## Introduction

Major international public health emergencies are the enemy of human life and health, which have a significant impact on regional and even global economy and society [1]. The COVID-19 epidemic in 2019 had a huge impact on the economy, society and people's lives and property, and seriously affected the physical, psychological and economic order of the public. Since the outbreak of COVID-19, Chinese government departments at all levels have promptly introduced a series of policies in response to the epidemic, including supporting epidemic prevention and control, stabilizing market order, regulating the macro-economy and avoiding social panic. From the perspective of policy, the epidemic prevention and control requires the

**Competing interests:** The authors have declared that no competing interests exist.

full cooperation of multiple departments to achieve epidemic prevention and control and social stability. As the embodiment of the governance ability, the policy directly reflects the governance level of the government, such as its policy thought, timeliness, implementation strength, etc.

Since SARS in 2003, public health emergency management has entered the field of view of researchers. In recent years, there are many relevant research achievements in China, mainly involving the emergency management system for colleges and communities [2], epidemiological characteristics of infectious diseases [3], management of online public opinion in major emergencies [4], survey and capacity evaluation of the emergency response capacity of disease prevention and control institutions and hospitals [5, 6]. However, there are few studies based on the perspective of policy tools, most of them focus on the policies of enterprises' resumption of work and production in the context of epidemic [7], and there is a lack of specific studies on epidemic prevention and control policies. Policy tools are the policy measures and means adopted by the government to achieve specific policy objectives, which reflect the ruling philosophy and value of the decision-maker [8]. Quantitative analysis of policy texts using policy tools is the mainstream policy analysis method at present, which has been widely used in e-commerce, agriculture, science and technology finance, pension, industrial Internet and other fields. Alexander Sokolov et al. used policy tools to conduct quantitative analysis of international science and policies on technology innovation cooperation to more rationally select a set of cooperation priorities for policy makers [9]. For a large-scale analysis of public health in Belgium, Jolien Grandia and Peter Kruyen used quantitative text analysis tools to assess the implementation of sustainable public procurement [10]. Ewalt Jag and Jennings Jr contributed to the welfare policy debate by examining the importance of specific policy instruments and the role in the dramatic decline in the number of welfare cases in public administration [11]. Willie J. Redmond used a computable equilibrium model to incorporate the impact of policy reforms into a policy analysis model to quantify the welfare effects of trade policies reformation in the Uruguay Round Negotiations on Agriculture [12]. Ben Williamson surveyed and mapped the landscape of digital policy instrumentation in education, and the digital policy instruments prefigure the emergence of 'real-time' and 'future-tense' techniques of digital education governance [13]. To study sustainable tourism management in Crikvenica, Croatia, Ivana Logar assessed the selected economic, regulatory and institutional policy instruments [14]. Timothy Carter and Laurie Fowler evaluated existing international and North American green roof policies at the federal, municipal, and community levels [15]. Hou Zhenxing and Lu Yan used policy tools to study the e-commerce policies of agricultural products [16]. Xiong Xiaogang used policy tools to analyze the content of China's entrepreneurship and innovation policy [17]. Wang Xiaofan, et al. studied China's cultivated land ecological management and protection policies through policy tools [18]. Huang Xinping et al. use policy tools to study China's technology and financial development policies [19], pension service policies [20], and industrial Internet policies respectively [21]. Peng Jisheng et al. proposed for the first time to construct quantitative indicators of innovation policy text from three dimensions of policy intensity, policy objectives and policy measures [22]. Later, Li Jinghua and Chang Xiaoran [23], Wang Bangjun and Zhu Rong [24], Zhang Yongan and Yan Jin [25] and others used the policy quantification method of Peng Jisheng et al. for reference. At present, the research of policy effectiveness evaluation in China is in the exploratory stage, the standards and methods of effectiveness evaluation are formed preliminarily.

Epidemic prevention policy is a systematic project of national response to complex epidemics, involving the integration of almost all regions, sectors, industries and resources. Policy can have maximum effect only when all parties, from the central government to local governments,

from the medical sector to support industries, fully cooperate. Therefore, scientific evaluation of epidemic policy has become an important part of policy performance.

The purpose of this study is to analyze how the Chinese government uses policy tools in the management of the epidemic crisis, how the policy tools change with the development of the epidemic, and the existing problems. In order to answer these questions, an analytical framework based on policy tools is constructed in this study to analyze the policies introduced by China during the COVID-19 epidemic, explore the rationality of the use of policy tools, analyze existing problems and put forward suggestions for optimization, so as to enrich the policy research on COVID-19.

The contributions of this study are as follows:

It promotes the classification research of policy tools in the management of major public health crises, and enriches the knowledge of the use of policy tools in the management of major public health crises. From the perspective of the relationship between standards, information and behavior, combined with the specific characteristics of COVID-19 policies, this study shows a combined analysis model of axial coding and selective coding of policy documents. The model includes the basic structure of COVID-19 control policy tools and the interaction between them, which provides a reference framework for the analysis of specific policy tool on COVID-19.

## Materials and methods

### Data acquisition and preparation

The national policies on COVID-19 from December 2019 to December 2021 were chosen as the research objects, and the policy texts issued by the Central Committee of the Communist Party of China, the State Council and its subordinate departments, the National Health Commission and other relevant departments are taken as the objective evidences. Collect policies through the following three channels: Firstly, relying on the State Council and its affiliated departments and other relevant official websites. Secondly, direct search in China Policy Net, Peking University Law Book, and law enforcement database. Thirdly, the relevant content in relevant literature and policy text is retrieved retrospectively.

In order to ensure that the content of policy information is consistent with the theme of national response to COVID-19, the following principles are followed when sorting and selecting policy texts: First, choose policies whose text is closely related to the prevention and control of COVID-19, do not count policies that only mention COVID-19 in the text. Secondly, the text type is the published general document administrative system, involving laws, regulations, provisions, decisions, plans, opinions, measures, notices, etc. Finally, 301 policy documents related to COVID-19 at the national level have been determined to build a database of COVID-19 policies.

### Statistical analysis

Using text-mining software ROSTCM6, 301 policies were imported into text-mining database, word segmentation was carried out on the document set, and then word frequency statistics were carried out, which were displayed successively from high frequency to low frequency.

In order to explore the level of operation of policy tool, based on cybernetics to study the policy tools in accordance with the framework of 'standards-information-behavior' [26]. Select different keywords from the basic elements of policy orientation to analyze the basic key elements of policy tools. After that, the main axes and selective codes of policy tools were formed by comparison and classification, and then the policy tools adopted by the government at different stages of epidemic development were analyzed.

## PMC-AE index model

In order to ensure the value and applicability of research results, eight representative policies were selected from 301 policy documents on COVID-19 collected (Table 1). The selection is followed three aspects. Firstly, as COVID-19 is a major national event, only policy texts published by the central government is selected, while those issued by local government departments are not included in the selection. Second, the principle of relevance. Select the policy texts that are most closely related to COVID-19 policies, such as 'epidemic prevention and control', 'resumption of work and production', 'stable employment' and 'supply chain recovery'. Third, the normative principle. In terms of policy effectiveness, give priority to legislative and administrative policies, including laws, regulations, decisions and opinions.

The Policy Modeling Consistency Index (PMC) is a policy text analysis model proposed by Ruiz based on the Omnia Mobilis hypothesis [27]. Ruiz believes that the variables in the world are interconnected and keep moving, and the influence of any one related variable cannot be ignored [28]. PMC index model can not only evaluate the consistency of policies, but also directly reflect the pros and cons of policies [29]. It is a relatively advanced policy evaluation model in the world at present [30]. The difference between PMC-AE index model and PMC index model lies in the method adopted to calculate the index. The former uses AE technology to fuse parameters, which is more advantageous than the linear fusion of the latter.

The steps of establishing the PMC-AE index model are as follows: Firstly, PMC index model is used to classify variables and identify parameters. Secondly, the multiple input-output table of COVID-19 policies is constructed, and text-mining technology is used to assign values. Then, the self-coding technology of neural network is used to fuse the multiple parameters, and the PMC-AE index is obtained, which is the score of each policy. Finally, the PMC-AE curve is drawn to evaluate the policies introduced under the impact of COVID-19 [31] (Fig 1).

In order to make a more targeted evaluation of China's epidemic prevention policies, 9 first-level variables and 36 second-level variables were established by referring to Ruiz Estrada [32] and scholars' evaluation variables of big data development policies in existing literatures

**Table 1. Eight representative policies.**

| Numbering | Policy name | Release date |
|---|---|---|
| P1 | The State Council: Do a good job in normalizing the prevention and control of the new crown pneumonia epidemic | May 7, 2020 |
| P2 | The State Council office: Released guidelines on the implementation of measures to stabilize employment in response to the impact of COVID-19 | March 18, 2020 |
| P3 | National Health Commission: Under the normalization of the prevention and control of the new crown pneumonia epidemic, the pre-hospital medical emergency response capacity will be further improved | July 9, 2020 |
| P4 | Medical Insurance Bureau: Cooperate with the work related to further improving the testing capacity of the new crown virus | June 16, 2020 |
| P5 | State-owned Assets Supervision and Administration Commission: Further do a good job in reducing the rent of small and micro enterprises and individual industrial and commercial households in the service industry | June 4, 2020 |
| P6 | The State Administration of Market Supervision and other 7 departments: Issued the "National Epidemic Prevention Materials Product Quality and Market Order Special Rectification Action Plan" | May 11, 2020 |
| P7 | The Ministry of Civil Affairs and National Health Commission: Issued the "Guidance Plan for the Precision and Refinement of Community Prevention and Control and Service Work of the New Crown Pneumonia Epidemic" | April 14, 2020 |
| P8 | State Administration of Taxation: Give full play to the role of tax functions to help win the battle against the epidemic | February 10, 2020 |

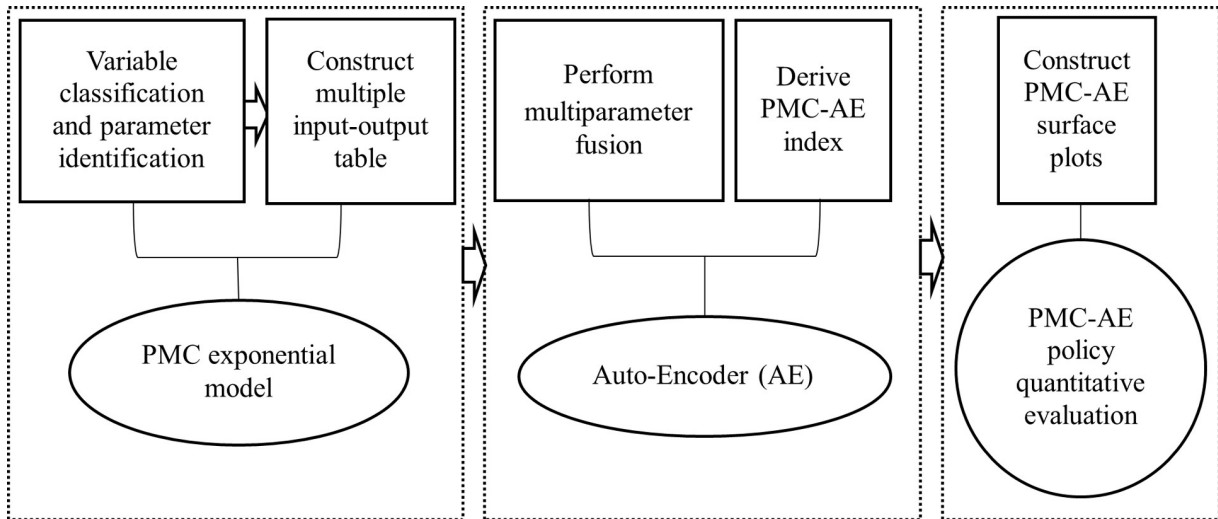

**Fig 1. PMC-AE index model construction process.**

and combined with the specific characteristics of COVID-19 policies [8, 33–37]. The specific variable design is shown in the following table (Table 2).

In this study, the text-mining method is used to determine the value of the second-level variables, and the ROSTCM6 software is used to import the policies into the text-mining database for word segmentation [38]. The value of the second-level variables is assigned according to the keywords of each policy text. When the policy text contains the corresponding keywords of the second-level variables, the value is 1; otherwise, it is 0.

**Table 2. Variable design of PMC evaluation index system.**

| Level-one variable | Level-two variable serial number | Level-two variable name | Level-two variable serial number | Level-two variable name |
|---|---|---|---|---|
| Nature of policy X1 | X1:1 | Forecast | X1:2 | Supervision |
| | X1:3 | Suggestion | X1:4 | Description |
| | X1:5 | Steer | | |
| Efficacy of policy X2 | X2:1 | Long term | X2:2 | Medium term |
| | X2:3 | Short term | | |
| Policy issuing agency X3 | X3:1 | State council | X3:2 | Ministries of the state |
| Policy evaluation X4 | X4:1 | Target specific | X4:2 | Scientific scheme |
| | X4:3 | Fully implemented | X4:4 | Rational planning |
| Policy domain X5 | X5:1 | Society | X5:2 | Enterprise |
| | X5:3 | Market | X5:4 | Livelihood |
| | X5:5 | Region | | |
| Incentive measures X6 | X6:1 | Government subsidies | X6:2 | Financial support |
| | X6:3 | Work instructions | X6:4 | Policies and regulations |
| | X6:5 | Division of duties | X6:6 | Tax deductions |
| Policy priorities X7 | X7:1 | Direct the work | X7:2 | Stable employment |
| | X7:3 | Corporate support | X7:4 | Rectify the market |
| | X7:5 | Serve the people's livelihood | | |
| Policy receptors X8 | X8:1 | Enterprise | X8:2 | Government |
| | X8:3 | Populace | X8:4 | Medical |
| Policy perspective X9 | X9:1 | Macro | X9:2 | Microcosmic |

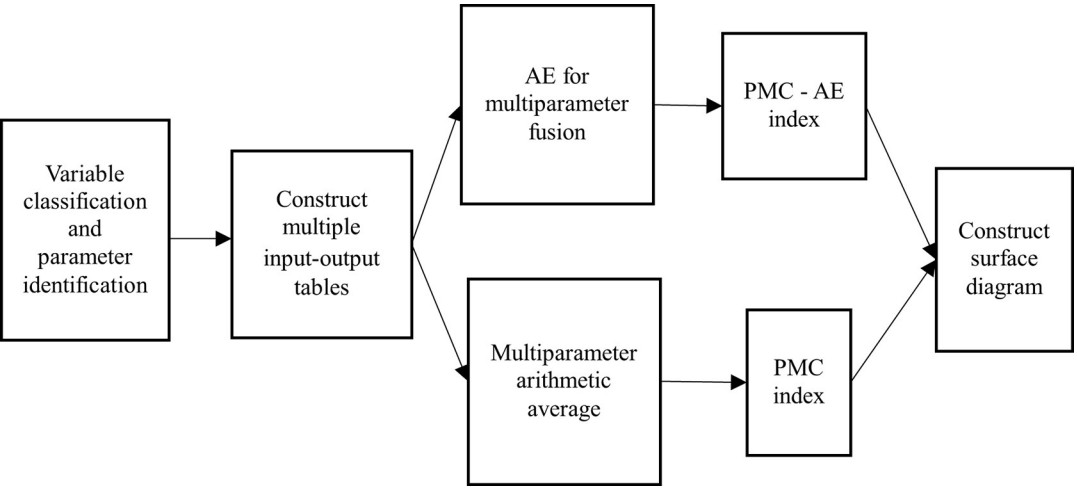

**Fig 2. Comparison of PMC-AE index model with PMC index model.**

Compared with expert scoring, text-mining method is more objective and scientific. Based on the above characteristics and variables of China's COVID-19 response policies, a multivariate input-output table of COVID-19 policies was obtained and values could be assigned to each policy according to the above methods.

Based on the PMC, the Auto Encoder technology (AE) is integrated in the neural network theory, and a quantitative evaluation tool of epidemic prevention and control policies is formed, that is PMC-AE index evaluation model, effectively avoiding the difficult to measure the relationship between policy indicators in the PMC index model. The comparison of PMC-AE index model with PMC index model is shown in the following figure (Fig 2).

Auto Encoder (AE) is a neural network with three or more layers (Fig 3), which belongs to unsupervised learning [39]. Specific learning process is: the minimum input nodes and output node differs as the purpose, first, nonlinear method to encode the original data are used to get the hidden layer nodes, and then to decode the hidden layer nodes is output layer nodes, the multiple cycles to study the optimal weights and constant input and output are minimal.

$$h = f(WX + b_1) \quad \text{hidden} \tag{1}$$

$$Y = g(Wh + b_2) \quad \text{output} \tag{2}$$

In the above formula $X = (x_1, x_2, \cdots, x_n)^T$ means the multi-dimensional epidemic prevention policy evaluation indicators established, $Y = (y_1, y_2, \cdots, y_n)^T$ means the corresponding output layer node value; f and g are activation functions for the hidden layer and output layer, commonly used Sigmoid, Tanh, Softplus functions, etc., f and g can be the same or different; $h = (h_1, h_2, \cdots, h_m)^T$ means hide the layer node value; $W$ and $W^T$ means the weight matrix between the input layer and the hidden layer and the weight matrix between the hidden layer and the output, usually the number of rows of the weight matrix is equal to the number of neuron nodes in the previous layer, and the number of columns is equal to the number of neuronal nodes in the next layer; $b_1 = (b_1, b_2, \cdots, b_m)^T$ and $b_2 = (b_1, b_2, \cdots, b_n)^T$ are respectively constant terms from the input layer to the hidden layer and from the hidden layer to the output layer, and the dimensions are the number of nodes corresponding to the next layer of

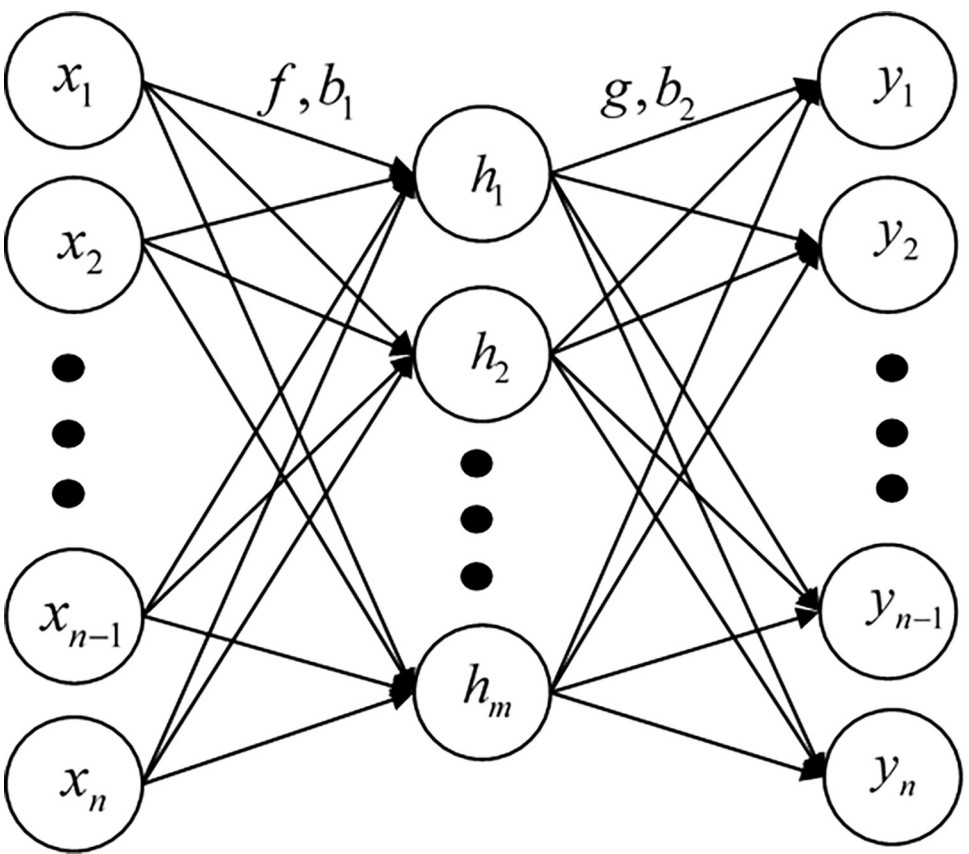

**Fig 3. Three-layer AE structure.**

neurons, respectively. The purpose of AE training is to make Y and X as similar as possible
[31].

$$W = \begin{bmatrix} W_{11} & W_{21} & \cdots & W_{m1} \\ W_{21} & W_{22} & \cdots & W_{2m} \\ \vdots & \vdots & \ddots & \vdots \\ W_{n1} & W_{n2} & \cdots & W_{nm} \end{bmatrix} \in R^{n \times m} \tag{3}$$

When the original data dimension is too high or data of low dimension needs to be
obtained, the number of layers of the neural network can be appropriately increased, and the
number of nodes of the neural unit can be reduced layer by layer. Therefore, after training, $X$
forms $H$ through nonlinear combination, and $H$ forms $Y$ through nonlinear combination, and
$Y = X$. Therefore, H is obtained by the integration of $X$ through nonlinear operation, while $Y$ is
obtained by decoding $h$. Therefore, $H$ can be considered as the nonlinear expression of $X$ and
$Y$, so $H$ can be used as the score of policy text after the integration of various indicators.

## Results

### Policy content keyword

Since the sample selected is the novel coronavirus policy issued by the State Council, words
such as 'General Office of the State Council', 'notice' and 'COVID-19' must have the highest

**Table 3. Statistics of valid vocabulary and word frequency in policy documents.**

| # | Vocabulary | Frequency | # | Vocabulary | Frequency | # | Vocabulary | Frequency |
|---|---|---|---|---|---|---|---|---|
| 1 | work resumption | 582 | 11 | government subsidies | 195 | 21 | finance | 172 |
| 2 | employment | 488 | 12 | epidemic situation reports | 194 | 22 | labor relation | 166 |
| 3 | goods and materials | 485 | 13 | management system | 191 | 23 | daily report system | 161 |
| 4 | detection | 388 | 14 | hygiene | 188 | 24 | psychological counseling | 160 |
| 5 | non-contact | 329 | 15 | medical resource | 182 | 25 | infection | 156 |
| 6 | medical staff | 329 | 16 | emergency management | 179 | 26 | service industry | 131 |
| 7 | infrastructure | 216 | 17 | logistics | 177 | 27 | transportation | 96 |
| 8 | regionalize | 212 | 18 | tax reduction and exemption | 177 | 28 | market normalization | 76 |
| 9 | finance support | 207 | 19 | approval process | 175 | 29 | public opinion guidance | 57 |
| 10 | quarantine | 203 | 20 | economics | 174 | 30 | market supervision | 43 |

frequency. According to the analysis of word frequency in ROSTCM6, words similar to the above have no significant affect policy classification and belong to redundant words. The adverbs of degree in high frequency words such as "major" and verbs such as "improve" also have no significant meaning to the content of policy texts, so they are redundant words and eliminated. After the deletion of redundant words, get the high-frequency words (Table 3).

As can be seen from the policy's keyword and word frequency policy, China's COVID-19 policy is mainly to provide economic support to enterprises and individuals affected by the epidemic, and to take emergency relief measures to prevent mass unemployment and bankruptcy caused by the pandemic crisis. Therefore, the keywords with high frequency are mainly 'resumption of work and production', 'employment', 'materials', 'financial support' and so on. Secondly, regarding the health crisis caused by the outbreak of COVID-19, the policy orientation adopted by China mainly focuses on technology promotion measures such as strengthening medical and health technologies, supplemented by closed measures such as 'isolation' and 'detection'. Public health-related personnel training, risk communication and mental health intervention also play a very important role in the prevention and control of COVID-19, so keywords such as 'medical staff', 'epidemic situation reports', 'daily report system' and 'psychological counseling' appear in the word frequency statistics. At the same time, word frequency statistics also reflect the impact of the epidemic on China's economy, health, society, services, transportation and other fields.

## Policy issuing department

China's epidemic prevention and control policies are issued by 49 departments, including the Joint prevention and Control working mechanism of the State Council and its ministries, the Supreme People's Court and the Supreme People's Procurator ate (Table 4). The top six departments issuing the most policies are: National Health Commission, Ministry of Finance, Ministry of Transport, Ministry of Commerce, Ministry of Human Resources and Social Security, and National Development and Reform Commission. This shows that epidemic prevention and control mainly involves health, finance, transportation, business, personnel, economic and social management and other major aspects. After the National Health Commission, the Ministry of Transport issued the most separate documents with 15. The Ministry of Finance made the joint announcements, with 24. In addition, the total joint mandate slightly more than independent departments issuing amount but this ratio is roughly equal, shows that in response to the outbreak, between the various departments of the Chinese government not only need to mobilize the field resources, also need coordination with other related departments, in order to make full use of its advantages in all departments, independent and joint

**Table 4. Policy issued department statistics.**

| Name | Quantity of articles published | Quantity of individually published | Quantity of jointly published |
|---|---|---|---|
| National Health Commission of the People's Republic of China | 63 | 35 | 35 |
| Ministry of Finance of the People′s Republic of China | 42 | 25 | 38 |
| Ministry of Transport of the People's Republic of China | 37 | 33 | 26 |
| Ministry of Commerce of the People's Republic of China | 30 | 27 | 14 |
| Ministry of Human Resources and Social Security of the People's Republic of China | 26 | 20 | 14 |
| National Development and Reform Commission | 24 | 17 | 12 |
| State Administration of Taxation | 21 | 15 | 10 |
| Ministry of Industry and Information Technology of the People's Republic of China | 20 | 13 | 9 |
| State Administration for Market Regulation | 18 | 11 | 6 |
| Ministry of Agriculture and Rural Affairs of the People's Republic of China | 16 | 10 | 5 |
| Working mechanism for joint prevention and control of the epidemic | 14 | 9 | 6 |
| Ministry of Civil Affairs of the People's Republic of China | 13 | 9 | 7 |
| National Administration of Traditional Chinese Medicine | 11 | 7 | 3 |
| The People's Bank of China | 9 | 7 | 6 |
| The Ministry of Education of the People's Republic of China | 8 | 7 | 9 |
| General Administration of Customs of the People's Republic of China | 7 | 7 | 5 |
| State Council of the People's Republic of China | 7 | 6 | 5 |
| China Banking Regulatory Commission | 7 | 5 | 4 |
| Ministry of Public Security of the People's Republic of China | 6 | 2 | 3 |
| Ministry of Ecology and Environment of the People's Republic of China | 5 | 4 | 3 |
| other departments | 5 | 38 | 42 |
| Total | 389 | 307 | 262 |

advantages to jointly cope with the epidemic challenge. The following table shows the policy making departments and the number of policies (Table 4).

## Application of policy tools

In terms of COVID-19 information collection and control, the main axis code consists of 'epidemic information submission' and 'epidemic information investigation'. The government pays special attention to keeping abreast of the latest situation by reporting local epidemic information and professional data. Based on mastering the epidemic situation, effective measures should be taken to control the epidemic. At the same time, internal investigations such as testing, infection distribution and close contacts should be carried out to ensure the safety of personnel and prevent further spread of the epidemic at the source.

On disease prevention and control measures, the axis is divided into 'non-structural measures' and 'structural measures'. Structural disaster relief, as the core content of disaster prevention and reduction and non-structural disaster reduction, can effectively ensure the safety of material supply, production flow and personnel during the epidemic. It shows the richness of the behavioral policy changes of the subjects related to disease prevention and control promoted by the Chinese government. In particular, traditional structural and non-structural measures form the core of epidemic prevention and control.

In the respect of the forms of epidemic intervention, the code mainly consists of 'emergency mobilization tool', 'joint support tool' and 'financial support tool'. Emergency mobilization

and epidemic prevention tools reflect the characteristics of passive risk management, that is, the "mobilization" and "rectification" caused by the incident, which is an important aspect of the Chinese government's risk management in response to emergencies. 'Services', 'Medical treatment', 'Materials' and 'Cooperation' constitute 'Joint protective epidemic prevention tools' as a form of organizational support. The tool of financial support is direct government assistance to small and medium-sized enterprises (Table 5).

According to the actual situation of China's epidemic prevention and control policy, the policy tools are divided into the strategic layer, the supply layer, the demand layer and the environmental layer by referring to relevant literature [40–44]. The strategic level refers to the government's medium and short-term plans for epidemic prevention, control and treatment, as well as the medium and long-term plans for emergency management, which play the role of top-level design and planning guidance. The supply layer refers to the prevention, control and treatment of the epidemic through measures in finance, technology, human resources, services and facilities. The demand layer refers to the government's control and restriction of community, transportation and market to reduce the epidemic rate. The environmental level refers to the regulatory measures put forward by the government for better epidemic prevention and control and the provision of a good social environment for epidemic prevention and control,

**Table 5. Axial and selective coding of COVID-19 subject policy documents.**

| Open coding | Axial coding | Selective coding |
|---|---|---|
| Report | Epidemic information report | Epidemic information control |
| Data | | |
| Collect | | |
| Detect | Epidemic information investigation | |
| Infect | | |
| Touch | | |
| Quarantine | Non-structural epidemic prevention measures | Epidemic control measures |
| Manage | | |
| Hedge | | |
| Supervise | | |
| Product | Structural epidemic prevention measures | |
| Work resumption | | |
| Logistics | | |
| Construction | | |
| Prevention and control | Emergency mobilization epidemic prevention tools | Forms of epidemic intervention |
| Community administration | | |
| Extend leave | | |
| In batches | | |
| Staggering peak | | |
| Service | Joint protective epidemic prevention tools | |
| Guarantee | | |
| Cooperation | | |
| Medical treatment | | |
| Goods and materials | | |
| Fiscal | Financial support for epidemic prevention tools | |
| Finance | | |
| Revenue | | |
| Rent | | |

**Table 6. Types, explanations and quantities of policy tools about epidemic prevention.**

| Classification | Tool name | Quantity | Percentage |
|---|---|---|---|
| Strategic layer | Planning scheme | 33 | 13.00% |
| Supply layer | Funding supply | 24 | 8.20% |
| | Manpower supply | 11 | 6.30% |
| | The scientific research support | 25 | 6.50% |
| | Public service | 14 | 11.70% |
| Demand layer | Safeguard | 13 | 10.30% |
| | Market regulation | 9 | 7.40% |
| | Channel constraints | 22 | 5.50% |
| | External exchange | 13 | 5.30% |
| Environment layer | Fiscal levy | 24 | 9.50% |
| | Targeted policies | 15 | 7.40% |
| | Governmental service | 9 | 6.60% |
| | Ecological environment | 2 | 2.30% |

that is, the role of providing guarantee. Based on the content analysis of 301 policies, the single times and total times of obtaining various policy tools are counted (Table 6).

It can be seen that the supply layer and the demand layer account for 32.7% and 28.5% respectively, the environmental layer accounts for 25.8%, and the strategic layer accounts for at least 13%. Moreover, government used a relatively large number of specific policy tools in dealing with the epidemic to ensure the full play of the effectiveness of all policies and achieve the effect of epidemic prevention and control and maintaining economic stability. Among them, the number and proportion of policies at the strategic level are more appropriate, their role is to make the policy system more systematic, and their number does not affect the priorities of other departments.

## Policy sample score

Based on the PMC-AE policy evaluation model constructed in this study, text-mining technology is used to score eight policies (Table 7).

Based on the score of eight policy texts, the neural network model was constructed with self-coding technology and parameters were learned. The process of data fusion consists of two stages. The first stage is to fuse the score of the second-level variables of each policy and obtain the score of the nine first-level variables of each policy. In the second stage, the scores of nine first-level variables were fused to obtain the PMC-AE index of each policy (Table 8).

The table shows that the calculation of PMC-AE index is not the traditional weighted average but the data fusion, and the value range of the result is not limited to 0–10. In addition, the higher the policy score is, the higher the level of the epidemic prevention policy is, and the policy coverage is wider.

As shown in Table 8, the policy score from high to low is P5>P1>P8>P7>P3>P2>P6>P4. Combined with the characteristics of the selected function, the higher the policy score, the higher the effectiveness of the COVID-19 policy and the wider the impact content. On the contrary, the level of COVID-19 prevention and control policies is relatively low, and there is still much room for improvement. The selected policy samples can be roughly divided into three grades according to their scores: (1) Policies whose score is between 12 and 14 are level I: P5, P1, P8 and P7. (2) Policies whose score is between 11 and 12 are categorized as level II: P3, P2 and P6. (3) Policy whose score is between 9 and 10 is classified as grade III: P4.

**Table 7. Multiple input-output table of eight policies.**

|  |  | P1 | P2 | P3 | P4 | P5 | P6 | P7 | P8 |
|---|---|---|---|---|---|---|---|---|---|
| X1 | X1:1 | 0 | 0 | 0 | 0 | 0 | 0 | 0 | 1 |
|  | X1:2 | 1 | 0 | 1 | 0 | 1 | 1 | 0 | 1 |
|  | X1:3 | 1 | 1 | 0 | 1 | 0 | 0 | 1 | 1 |
|  | X1:4 | 1 | 1 | 1 | 0 | 1 | 1 | 1 | 0 |
|  | X1:5 | 1 | 1 | 1 | 1 | 1 | 0 | 1 | 1 |
| X2 | X2:1 | 1 | 0 | 1 | 1 | 0 | 1 | 0 | 0 |
|  | X2:2 | 1 | 1 | 1 | 1 | 0 | 1 | 1 | 1 |
|  | X2:3 | 0 | 1 | 0 | 0 | 1 | 0 | 1 | 1 |
| X3 | X3:1 | 1 | 1 | 0 | 0 | 0 | 0 | 0 | 0 |
|  | X3:2 | 0 | 0 | 1 | 1 | 1 | 1 | 1 | 1 |
| X4 | X4:1 | 1 | 1 | 1 | 1 | 1 | 1 | 1 | 1 |
|  | X4:2 | 1 | 1 | 1 | 1 | 1 | 1 | 1 | 0 |
|  | X4:3 | 1 | 0 | 1 | 0 | 1 | 0 | 0 | 0 |
|  | X4:4 | 1 | 1 | 0 | 1 | 1 | 1 | 1 | 1 |
| X5 | X5:1 | 1 | 1 | 0 | 1 | 0 | 0 | 0 | 1 |
|  | X5:2 | 0 | 1 | 0 | 0 | 1 | 0 | 0 | 1 |
|  | X5:3 | 0 | 0 | 0 | 0 | 0 | 1 | 0 | 0 |
|  | X5:4 | 1 | 1 | 1 | 0 | 1 | 0 | 0 | 1 |
|  | X5:5 | 1 | 0 | 1 | 0 | 0 | 0 | 1 | 0 |
| X6 | X6:1 | 0 | 0 | 0 | 0 | 1 | 0 | 0 | 0 |
|  | X6:2 | 0 | 0 | 0 | 0 | 1 | 0 | 0 | 0 |
|  | X6:3 | 1 | 0 | 1 | 1 | 0 | 1 | 1 | 1 |
|  | X6:4 | 0 | 1 | 0 | 0 | 1 | 0 | 0 | 0 |
|  | X6:5 | 1 | 1 | 1 | 0 | 0 | 1 | 1 | 0 |
|  | X6:6 | 0 | 0 | 0 | 0 | 1 | 0 | 0 | 1 |
| X7 | X7:1 | 1 | 0 | 1 | 1 | 0 | 0 | 1 | 1 |
|  | X7:2 | 0 | 1 | 0 | 0 | 0 | 0 | 0 | 0 |
|  | X7:3 | 0 | 0 | 0 | 0 | 1 | 0 | 0 | 0 |
|  | X7:4 | 0 | 0 | 0 | 1 | 0 | 1 | 0 | 0 |
|  | X7:5 | 0 | 0 | 0 | 0 | 1 | 0 | 1 | 1 |
| X8 | X8:1 | 0 | 0 | 0 | 0 | 1 | 0 | 0 | 1 |
|  | X8:2 | 1 | 0 | 0 | 0 | 0 | 0 | 0 | 0 |
|  | X8:3 | 1 | 1 | 0 | 0 | 1 | 1 | 1 | 1 |
|  | X8:4 | 0 | 0 | 1 | 1 | 0 | 0 | 0 | 0 |
| X9 | X9:1 | 1 | 1 | 1 | 0 | 0 | 1 | 0 | 1 |
|  | X9:2 | 0 | 0 | 0 | 1 | 1 | 0 | 1 | 0 |

Policy sample P5, 'Further do a good job in reducing the rent of small and micro enterprises and individual industrial and commercial households in the service industry', is a direct policy tool for the service sector, small and micro enterprises, industrial and commercial enterprises that have been hit hard by the epidemic. After the policy was released, all provinces and cities responded positively and gave feedback on the implementation. This policy has a clear division of responsibilities and a monitoring and inspection mechanism, which shows that this policy is excellent.

The score of policy P1, 'Do a good job in normalizing the prevention and control of the new crown pneumonia epidemic', is in the second place. By observing the surface graph of P1 and P8 and comparing the scores of first-order variables in the policy-scoring table, it can be

**Table 8. Policy sample score situation.**

|  | X1 | X2 | X3 | X4 | X5 | X6 | X7 | X8 | X9 | Final score |
|---|---|---|---|---|---|---|---|---|---|---|
| **P1** | 2.673 | 0.856 | 0.745 | 2.764 | 0.898 | 2.733 | 0.536 | 1.638 | 0.764 | 13.607 |
| **P2** | 2.576 | 0.856 | 0.745 | 1.845 | 0.753 | 2.639 | 0.554 | 0.745 | 0.764 | 11.477 |
| **P3** | 2.587 | 0.856 | 0.670 | 2.378 | 0.676 | 2.733 | 0.536 | 0.636 | 0.764 | 11.836 |
| **P4** | 2.494 | 0.856 | 0.670 | 1.845 | 0.528 | 2.565 | 0.632 | 0.636 | 0.687 | 10.913 |
| **P5** | 2.543 | 0.464 | 0.670 | 2.764 | 0.715 | 3.237 | 0.746 | 1.936 | 0.687 | 13.762 |
| **P6** | 2.543 | 0.856 | 0.670 | 1.845 | 0.547 | 2.639 | 0.537 | 0.745 | 0.764 | 11.146 |
| **P7** | 2.576 | 0.856 | 0.670 | 1.845 | 0.504 | 2.639 | 1.357 | 0.745 | 0.687 | 11.879 |
| **P8** | 2.613 | 0.856 | 0.670 | 0.934 | 0.753 | 2.565 | 1.357 | 1.936 | 0.764 | 12.448 |
| **mean** | 2.576 | 0.807 | 0.689 | 2.028 | 0.672 | 2.719 | 0.782 | 1.127 | 0.735 | 12.134 |

found that the difference mainly lies in the two first-order variables of policy evaluation X4 and policy focus X7.

Policy score depends on policy evaluation. Policy samples P2, P3 and P6 help balance the relationship between epidemic prevention and control and society, economy, market, enterprises and people's livelihood. The final scores of P2, P3 and P6 show little difference, but compared with those of policy samples P1, P5, P7 and P8, there is still room for improvement.

Policy P8, 'Giving Full play to tax Functions to Help Win the Battle of Epidemic Prevention and Control', ranks the third. It is a financially supportive outbreak control tool designed to minimize the costs of a major public health crisis through economic means. In fact, it is also government intervention in the market during the crisis. The policy focuses on actively adjusting tax administration measures, and companies affected by the epidemic can defer tax payments. In particular, small and micro enterprises with serious difficulties in production and operation due to the epidemic can enjoy a special tax extension policy, to ease their financial pressure. Thus promoting the resumption of work and production of small and medium-sized enterprises and economic development. In order to maintain GDP growth during the regular epidemic prevention and control period, China further increases tax and fee cuts based on the tax multiplier effect, to increase residents' disposable income, stimulate consumption and promote economic growth. Therefore, the policy objectives of this policy are relatively clear, especially to simplify the examination and approval process and non-contact handling to further implement the tax and fee reduction policies and measures. However, this policy lacks pilot demonstration and macro-level control, resulting in the score of this policy sample in policy evaluation index is lower than the average.

Policy P7, 'COVID-19 outbreak community prevention, control, and the service work for the fine guidance precision', is a grass-roots management policy. In China's epidemic prevention and control system, communities are the most basic unit, taking on the arduous task of preventing the spread of the epidemic, ensuring people's livelihood and facilitating the resumption of work and production, thus making important contributions to China's epidemic prevention and control work. However, the score of this policy in policy field X5 and policy receptor X8 is lower than the average level, indicating that this policy is highly targeted and belongs to the specific epidemic prevention measures. In general, this policy is very important from the perspective of micro control.

There is little difference in the policy sample P3, P2 and P6 scores. The low score of P2 in policy evaluation is due to the low score of policy implementation. Local governments have introduced active measures to promote employment, especially a series of employment plans for college graduates. Nevertheless, in the economic downturn, hiring plans are shrinking. Among them, private enterprises and some foreign-funded enterprises faced the impact of the epidemic and even experienced large-scale layoffs.

P3 and P6 are the supervision policies for emergency treatment outside hospitals and epidemic prevention materials, which belong to the joint guarantee of epidemic prevention tools. The two policies provide equipment and materials to protect people's lives during the epidemic, and they are branches of the epidemic prevention system. Its low score lies in its poor pertinence, small audience and lack of collaboration.

Policy sample P4, 'To further improve the impact of coronavirus detection capability on work', receives the lowest score. The policy not only target period is uncertain, but also the participants have limitations, and lack of specific operation and supervision mechanism. Therefore, it can be improved in the future from the aspects of the object of participation and the nature of the policy. The specific improvement path is suggested as follows: X9-X5-X8-X4.

## PMC-AE surface

According to the score of the policy sample, the PMC-AE surface chart of the policy can be drawn, that is, the score of the integrated first-level variables in the policy sample can be converted into a third-order matrix, and the specific value of the third-order matrix can be calculated by the following formula (Fig 4).

$$P = \begin{pmatrix} X1 & X2 & X3 \\ X4 & X5 & X6 \\ X7 & X8 & X9 \end{pmatrix} \tag{4}$$

As can be seen from the figure above, the area of the surface diagram of policy samples P1, P5 and P8 is significantly larger than that of policy samples P2, P3, P4, P6 and P7. It can be seen from Table 8 that the score of these three policy samples is higher than that of the other five. Moreover, the scores are all high, indicating that the text content of the evaluated policies is relatively good.

The overall changes in the three COVID-19 policy scores can be visually observed in Fig 5. The indexes that changed greatly are mainly reflected in policy effectiveness X2, policy evaluation X4 and incentive measures X7. The indexes with relatively little change are policy area X5, policy focus X6, and policy receptor X8. Unchanged are the policy nature X1, issuing agency X3 and Policy Perspective X9.

## Discussion

It can be seen from the study that China's COVID-19 prevention and control policies aim to achieve effective management of major public health crises. The strategy is a dynamic match between the public health crisis management objectives at different stages of COVID-19 and a variety of policy tools. The research on the scientific combination and intrinsic properties of different policy tools in the process of COVID-19 epidemic management in China has the characteristics of typical cases. In fact, in the international comparison of COVID-19 response, the typical features of China's crisis management are that the phased objectives, the combination of policy tools and the level of application are well adapted. Therefore, the research of this paper has important reference significance for other crisis management in China and other countries. China's COVID-19 prevention and control policies have scientific features. The objectives, division of labor, responsibility and mechanism of the policy are clear. All policy tools reflect efforts to manage the relationship between epidemic prevention and control, social, economic, market, business and people's livelihood. China has adopted direct policy tools for small and micro enterprises and individual businesses that are experiencing

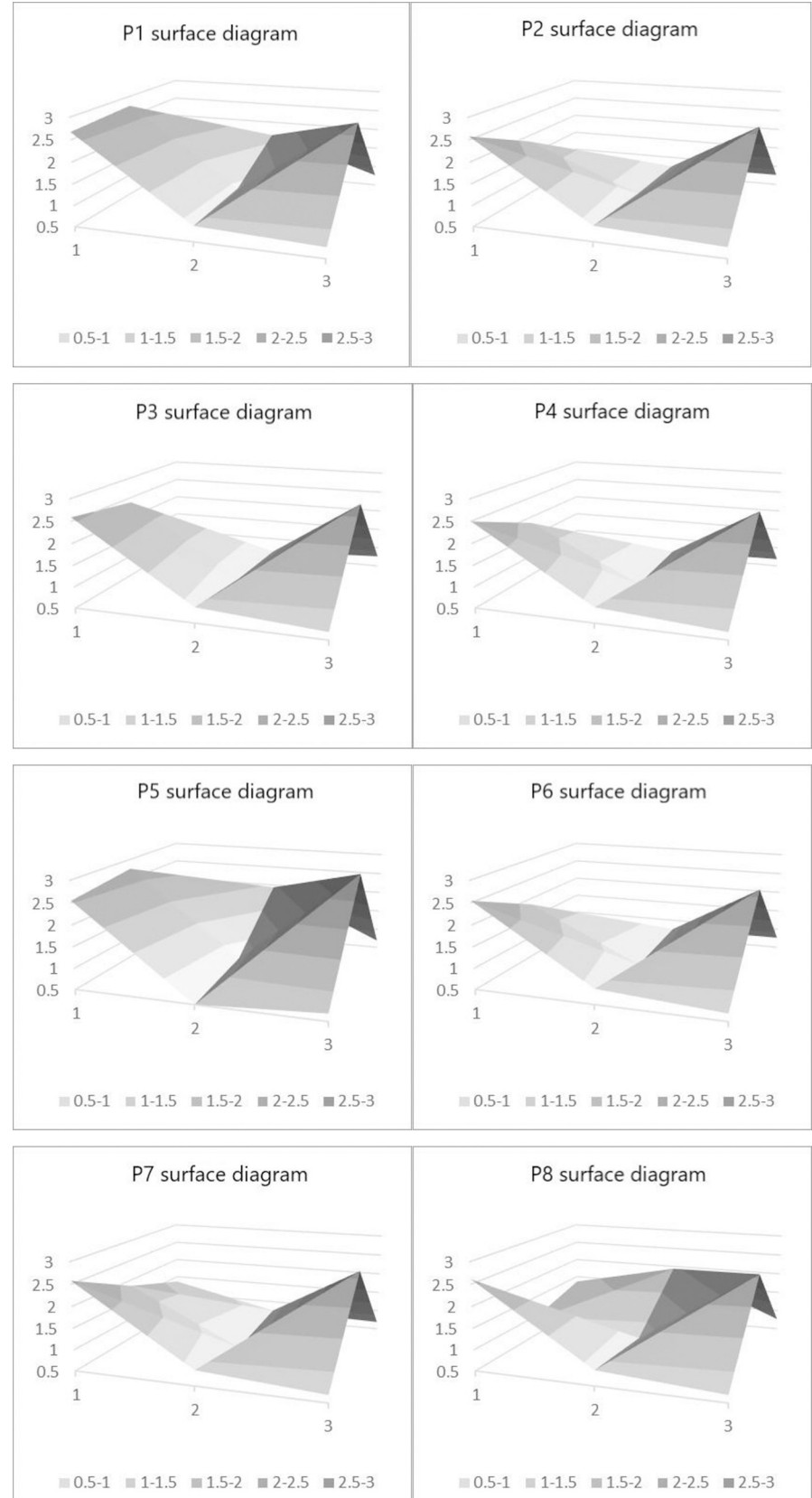

**Fig 4. PMC-AE surface diagram of each policy.**

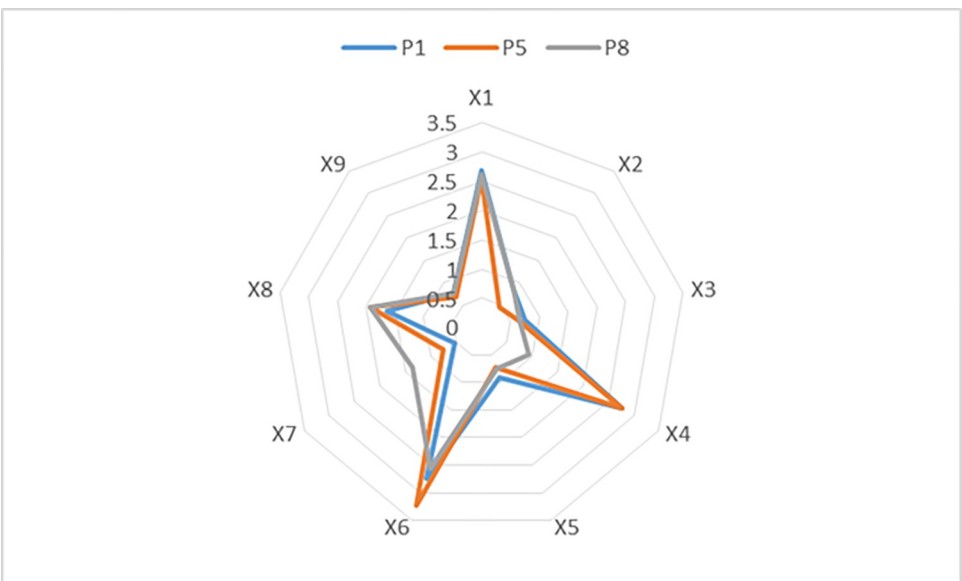

**Fig 5. The radar maps of three COVID-19 policies.**

difficulties because of the impact of the pandemic. After the policy was released, all provinces and cities responded positively and gave timely feedback on the implementation of the policy.

China's epidemic prevention and control policies attach importance to medical and preventive measures, which have played an important role in controlling the spread of the epidemic in China. Strict social distancing is also very costly in terms of economical and psychological damage, which naturally leads to a multi-objective decision problem. This is in line with a series of epidemic prevention and control policies issued by China, such as quarantine, no contact, psychological counseling and other policies. With the development of the epidemic, keeping the lockdown policy strict can never lead to a stable equilibrium when ending the lock down, no matter how long it did take place before all measures were suspended [45–48]. Hence, many countries have even already started to loosen the lockdown in very small steps [49]. China has also introduced normalized management, and the research of this paper shows that this policy sample has a high score and involves a wide range of contents. Overall, China's policy mix is consistent with the development pattern of the epidemic found in existing studies.

The main problem with China's use of epidemic prevention and control policy tools is that financial tools are not effective. Such policies have clear policy objectives, but lack of pilot demonstration and macro-level control. Targeted epidemic prevention measures are highly targeted, but such policies are limited in policy areas and policy recipients. The joint epidemic prevention tools mainly provide equipment and materials for the life and safety of people during the epidemic. However, such policies' scores are low in the policy field, mainly due to their strong targeting, small audience and lack of uniform standards, which are caused by the characteristics of the policies themselves. Some research on COVID-19 policy theme mining and hierarchical diffusion characteristics analysis based on spatio-temporal big data evolution also reflects this problem [50].

The timing of the outbreak of the COVID-19 crisis varies among different regions in China, which may lead to differences in crisis response stages and tools. This study starts with the policy tools at the central level and takes the epidemic crisis management in China as a whole, ignoring the differences among regions to some extent. At the same time, the

application of policy instruments requires the efforts of multi-party cooperation, which is usually not limited to the activities of policy implementation organizations, but also involves the interaction between implementers and actors in the policy implementation environment [51–53]. More systematic studies of different regions with public health policy tools could be considered in the future. The systematic study of public policy tools is helpful to promote the discipline construction of public policy analysis, and will provide strong methodological and theoretical support for the design and implementation of public policy tools in different disciplines and social practice fields [54].

## Conclusion

This study demonstrates the text-mining method to conduct a multi-dimensional exploration of the policies issued by China's central government since the outbreak of the novel coronavirus pneumonia, and comprehensively analyze the characteristics of various policy tools. Then, based on Omnia Mobilis hypothesis and neural network theory, combined with multiple input-output table in PMC index model and AE technology in data fusion theory, the PMC-AE evaluation model of COVID-19 policy was established.

Using text-mining software ROSTCM6, 301 strategies were imported into text-mining database. The analysis of key words shows that China's COVID-19 policies are mainly aimed at providing economic support to enterprises and individuals affected by the epidemic. The analysis of policymaking department indicates that 49 departments, including the Joint prevention and Control working mechanism of the State Council and its ministries, issue China's epidemic prevention and control policies and commissions, the Supreme People's Court and the Supreme People's Procurator ate. The analysis of policy tools makes clear that China's COVID-19 prevention and control policies include 32.7 percent supply-level and 28.5 percent demand-level, and 25.8 percent environment-level. In addition, strategy-level policies accounted for at least 13 percent.

According to the principle of openness, authority, relevance and normative principle, eight COVID-19 policies was chosen and quantitatively evaluated. The evaluation results show that: first, the eight policies can be divided into three grades, among which four policies are level I policies, three policies are level II policies and one policy is level III policies. The reason for the low scores of policies is that they are affected by four indicators: policy evaluation, incentive measures, policy emphasis and policy receptor. Second, COVID-19 policy design is characterized by 'top-down', and there is a large gap between policies at different levels. The overall scores of policies issued by the State Council are high, while the scores of policies issued by Chinese ministries and commissions are uneven, and the scores of specific epidemic prevention policies are relatively low.

In general, China's policies related to COVID-19 have focused on providing economic support to companies and individuals affected by the epidemic, as well as emergency relief measures to prevent mass unemployment and bankruptcies caused by the crisis. Secondly, in the health crisis caused by the outbreak of COVID-19, the policy orientation adopted by China is mainly technology promotion measures such as strengthening medical and health technology, supplemented by closed measures. Public health-related personnel training, risk communication and mental health interventions have played an important role in COVID-19 prevention and control. Further analysis shows that China has realized the whole process of epidemic prevention and control, including the control of epidemic information from the source, and adopted non-structural and structural epidemic prevention measures to control the epidemic after mastering the epidemic information. Finally, complicated intervention was carried out by issuing some specific and targeted epidemic prevention policies.

## Supporting information

**S1 Appendix. 301 original policy documents.**
(DOCX)

**S2 Appendix. List of policy names.**
(XLSX)

## Author Contributions

**Conceptualization:** Jianzhao Liu.

**Data curation:** Na Li.

**Formal analysis:** Jianzhao Liu.

**Methodology:** Na Li.

**Project administration:** Jianzhao Liu.

**Resources:** Jianzhao Liu.

**Software:** Na Li.

**Supervision:** Jianzhao Liu.

**Validation:** Luming Cheng.

**Visualization:** Na Li, Luming Cheng.

**Writing – original draft:** Na Li, Luming Cheng.

**Writing – review & editing:** Luming Cheng.

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
