## [Decision Letter · Decision Letter 0]

2 Sep 2022

PONE-D-22-20862Mining and quantitative evaluation of COVID-19 policy tools in ChinaPLOS ONE

Dear Dr. Cheng,

Thank you for submitting your manuscript to PLOS ONE. After careful consideration, we feel that it has merit but does not fully meet PLOS ONE’s publication criteria as it currently stands. Therefore, we invite you to submit a revised version of the manuscript that addresses the points raised during the review process.

We look forward to receiving your revised manuscript.

Kind regards,

Muhammad Mohiuddin

Academic Editor

PLOS ONE

Journal Requirements:

Additional Editor Comments:

Dear Authors,

We have received the reviewer's assessment of your paper. please reply to reviewer's comments. Please inform us if you can not reply to a queries that reviewers mentioned. I believe the quality of your paper will be improved if you can address the comments provided by our respective reviewers.

Thank you,

MM.

Reviewers' comments:

Reviewer's Responses to Questions

**Comments to the Author**

1. Is the manuscript technically sound, and do the data support the conclusions?

Reviewer #1: Yes

Reviewer #2: Yes

Reviewer #3: Yes

2. Has the statistical analysis been performed appropriately and rigorously? 

Reviewer #1: Yes

Reviewer #2: Yes

Reviewer #3: Yes

3. Have the authors made all data underlying the findings in their manuscript fully available?

Reviewer #1: Yes

Reviewer #2: Yes

Reviewer #3: Yes

4. Is the manuscript presented in an intelligible fashion and written in standard English?

Reviewer #1: No

Reviewer #2: Yes

Reviewer #3: Yes

5. Review Comments to the Author

Reviewer #1: Comments

1. The Abstract introduces a concept of BMC-AE. Even though it is in the Abstract, it should be explained, what it means. One sentence should suffice!

2. The sentence “finally, complicated intervention was carried out by issuing some specific and targeted epidemic prevention policies” in the Abstract should its’ own sentence.

3. The word “Control” in the Introduction section should npt be capitalized.

4. You said in the Introduction section that “Quantitative analysis of policy texts using policy tools is the mainstream policy analysis method at present, which has been widely used in e-commerce, agriculture, science and technology finance, pension, industrial Internet and other fields”. All references after that refer to studies done in China. Are there any international references that you could cite?

5. In “1.Policy tool analysis” heading there should be a space after the period!

6. In Table 1 you could replace the “number” with “#” to save space.

7. You mention in the “1.Policy tool analysis” that a text mining software was used. Which one?

8. In section “Policy tools analysis” you said that “ should be eliminated”. I would recommend using an even stronger expression of “were eliminated” as this is what actually happened?

9. Please the exact expression “COVID-19” consistently throughout the paper.

10. In section “1.2 Policy release department analysis” the word “Joint” should not be capitalized?

11. In section “1.2 Policy release department analysis”, should all the words in the expression “National health committee” be capitalized?

12. In Tables I would recommend to left align the first column!

13. In section “1.3 Policy tool analysis” to first sentence is confusing! Maybe it should be written as “…to clearly explore…”?

14. Should the column headings in the Tables be capitalized. Now, there is no consistency!

15. In Tables, should all the words in column 1 be capitalized? Same comment for the column headings!

16. After Table 3 "to clear to explore you wrote “and taking effective measures to control and control the epidemic on the basis of understanding the development trend of the epidemic.” The use of the word “control” in the sentence causes confusion. Maybe you could rephrase?

17. Before Table 4, you wrote “Financially supported epidemic prevention tools are direct government assistance for small and medium-sized enterprises.” As you use the semicolon before, should the word “Financially” be de-capitalized?

18. In the next paragraph you wrote “we divided policy tools into strategic layers, supply layers, demand layers and environmental layers”. Usually in academic papers, the use of words like “we”, and “I” should be avoided, and instead passive expression should be used consistently throughout the paper!

19. In section 2, you again introduce the expression PMC-AE. I think it would be a good idea to explain briefly what it in fact is!

20. The word “Secondly,” should be de-capitalized in the first paragraph of section 2.

21. Is there is Methodology section in this paper?

22. The words in the Tables are sometimes capitalized and sometimes not. There should be consistency!

23. In the last paragraph of section 2.1. you said “Compared with subjective expert scoring..” Should you rather say “Compared with subject expert scoring..”?

24. Should the figure headings include the complete word “Figure”?

25. In 2.3. you introduce the acronym Omnia Mobilis? There is no reference and there is no explanation. What is it?

26. In the same sentence you use the word “correlation”. Maybe a better word is relationship?

27. AE technology? Reference needed!

28. Figure 3 has been obviously copied from somewhere and therefore needs a reference. Also, it is better to draw the image again, as the copy is incomplete.

29. The first sentence after figure 3 needs a period at the end.

30. The paragraph after Figure 3 needs references.

31. The footnote font in text after Table 7 is not consistent with the font in the rest of the text.

32. The images in Figure 4 are unclear.

33. In section 3.4. the sentence “Policies with scores between 9 and 10 are classified as grade Ⅲ policies with P4” needs a period before it.

34. In section 3.4. second paragraph what is “sasac”?

35. The sentence “The score of policy P1 is in the second place” should start a new paragraph!

36. In section 3.4. the sentence “The differences in the final scores of policy samples P2, P3 and P6 are not significant”. Are you referring to statistical significance? If not, maybe a different word should be used?

37. Alos, in the same sentence you use the word “they”. To what exactly are you referring to? The scores? If this is the case, it needs to be said specifically!

38. The sentence “Policy P8 "Giving Full play to tax Functions to Help Win the Battle of Epidemic Prevention and Control" ranks the third” needs to start a new paragraph!

39. Maybe the sentence “Tax payment has been deferred in accordance with the law..” can be written as “tax payment can be deferred in accordance with the law..”

40. What are smes? I think you mean SMEs, and if so, that is the specific acronym to be used.

41. The sentence “Policy, sample P7 COVID - 19 outbreak community prevention and control and the service work for..” needs to start a new paragraph.

42. The font used in the references section differs for the rest of the text.

Reviewer #2: Thank you for giving is interesting from the topic to the findings. However, in its nature, I am unable to accept the manuscript. Some improvements are needed especially in the conclusion section. You will need to improve it based on the following comments.

a) Introduction

i) The objective of the study has not been clearly stated. It is advised that in the introduction, terms like “the objective of this study……” to be included followed by the study objectives

ii) I believe that “introduction” is the first section of the paper. Therefore, numbering should start at the introduction section named as follows:

1. Introduction

iii) The introduction is well written to introduce the topic. However, it is very shallow and small in size. It is advised to expand the introduction to cover all-important aspects of the study

b) Policy tools analysis

i) The policy analysis tool section is well discussed and presented. However, Fig 1 named “PMC-AE index model construction process” is not clearly visible. Redraw or enlarge the graph so that it can be clearly visible

ii) Figure 3. “Three-layer AE structure” is also blurred. Redraw the graph so that it is clearly presentable

c) Conclusions

i) The conclusion of the study is very shallow. There is need to rework out the conclusion and expand it with reference to the findings of the study

ii) The discussion section of the findings of the study is not presented. This section need to be on its own named as “discussion of the findings”

iii) The discussion section should discuss the findings of the study with respect to:

• The research objectives

• Research hypothesis – state whether the stated research hypothesis were confirmed or not

• The findings of the previous studies – state whether the findings of the study is in line with previous research findings or not

• The discussion section should reference the previous findings with in-text citation. Please consider citing the following studies.

Hoque, A., Mohiuddin, M., & Su, Z. (2018). Effects of Industrial Operations on Socio-Environmental and Public Health Degradation: Evidence from a Least Developing Country (LDC). Sustainability, 10(11), 3948. https://doi.org/10.3390/su10113948

Chaveesuk, S., Khalid, B., & Chaiyasoonthorn, W. (2022). Continuance intention to use digital payments in mitigating the spread of COVID-19 virus. International Journal of Data and Network Science, 6(2), 527–536. https://doi.org/10.5267/j.ijdns.2021.12.001

Khalid, B., Urbański, M., Kowalska-Sudyka, M., Wysłocka, E., & Piontek, B. (2021). Evaluating Consumers’ Adoption of Renewable Energy. Energies, 14(21), 7138. https://doi.org/10.3390/en14217138

iv) The paper should have the following section:

1. The “policy recommendation/implications” section – the section should discuss both the managerial implications as well as theoretical implications of the study.

2. Recommendations for future studies

d) General comments

i) The paper needs to be proofread and formatted correctly. The fonts of the content are different from that of the references

ii) The font sizes should also be standardized.

iii) The grammar should also be improved

Reviewer #3: The paper is interesting but it requires minor revision. The comments are given below:

1. Define the theoretical and practical implications as a separate section.

2. Define the limitations and future research after the conclusion section.

6. PLOS authors have the option to publish the peer review history of their article (what does this mean?). If published, this will include your full peer review and any attached files.

Reviewer #1: No

Reviewer #2: No

Reviewer #3: No

---

## [Author Response · Author response to Decision Letter 0]

5 Oct 2022

Dear Reviewers,

Thank you very much for your comments and suggestions on the paper "Mining and quantitative evaluation of COVID-19 policy tools in China". I will reply to each reviewer's comments one by one.

Many thanks to reviewer 1 for pointing out detailed errors in the grammar and format of the article. I have corrected these errors. In addition, I have also improved the supplement of references. In response to your question "Is there Methodology section in this paper?" My answer is that there is no separate section in the paper to discuss the research method. The third section of the paper, "Construction of PMC-AE Index Model for COVID-19 Policy Quantification", introduces the PMC-AE method，and applied it directly.

Thanks to reviewer 2 for your recognition of the topic of the paper. I have corrected the problems you raised, especially the introduction and conclusion. I include a discussion section to discuss how China has used a mix of policy instruments, how this mix has changed over the course of the crisis, and to compare it with previous studies. The questions raised in the introduction section are answered. What's more, the theoretical and practical implications of the study and recommendations for future studies are added in the introduction and discussion sections, respectively.

Thanks to reviewer 3 for your comments, which were few but useful. With your suggestion, I have added the theoretical and practical significance to the introduction section, and added the limitations of the article and future research direction of the article in the newly added discussion section.

Finally, I would like to express my sincere thanks again. I hope that you can put forward valuable comments on my revised article again and look forward to your reply.

Sincerely,

Luming Cheng

---

## [Decision Letter · Decision Letter 1]

30 Jan 2023

PONE-D-22-20862R1Mining and quantitative evaluation of COVID-19 policy tools in ChinaPLOS ONE

Dear Dr. Cheng,

Thank you for submitting your manuscript to PLOS ONE. After careful consideration, we feel that it has merit but does not fully meet PLOS ONE’s publication criteria as it currently stands. Therefore, we invite you to submit a revised version of the manuscript that addresses the points raised during the review process.

We look forward to receiving your revised manuscript.

Kind regards,

Peyman Rezaei-Hachesu, Associate professor

Academic Editor

PLOS ONE

Journal Requirements:

Reviewers' comments:

Reviewer's Responses to Questions

**Comments to the Author**

1. If the authors have adequately addressed your comments raised in a previous round of review and you feel that this manuscript is now acceptable for publication, you may indicate that here to bypass the “Comments to the Author” section, enter your conflict of interest statement in the “Confidential to Editor” section, and submit your "Accept" recommendation.

Reviewer #1: (No Response)

Reviewer #4: (No Response)

2. Is the manuscript technically sound, and do the data support the conclusions?

Reviewer #1: Yes

Reviewer #4: Yes

3. Has the statistical analysis been performed appropriately and rigorously? 

Reviewer #1: N/A

Reviewer #4: N/A

4. Have the authors made all data underlying the findings in their manuscript fully available?

Reviewer #1: No

Reviewer #4: Yes

5. Is the manuscript presented in an intelligible fashion and written in standard English?

Reviewer #1: Yes

Reviewer #4: No

6. Review Comments to the Author

Reviewer #1: Comments

1. The Abstract introduces the concept of BMC-AE. Even though it is in the Abstract, what should be explained, what it means? One sentence should suffice!

2. Table 3, the first columns should be left-aligned. Please check other tables as well.

3. In section 2, you again introduce the expression PMC-AE. I think it would be a good idea to explain briefly what it in fact is!

4. Is there is Methodology section in this paper, this was my original question. Every research paper should have this section.

5. Please widen the width of Table 7.

Reviewer #4: Dear Editor,

I am very pleased to have the opportunity of reviewing the revised version of the manuscript submitted to PLOS ONE Journal. The authors have addressed most of the comments made by the reviewers. However, the paper still needs major improvement.

Comments for authors:

- In general: English grammar is sometimes incorrect. The manuscript should be refined for English grammatical structure and phraseology.

- The abstract needs rework to highlight the main findings and conclusion.

- The paper is not well written. I have the impression that the presentation of the manuscript is confusing. Please provide separate sections for the method and result.

- Present the steps of the method (i.e., study design) clearly. I suggest that you also use Figure

- The authors should insert the "data acquisition and preparation" section in the method.

- What is the authors' method for text mining preprocessing?

- What is the authors’ method for selecting the official policy documents? Please clarify data collection and selection (e.g., inclusion criteria).

- What is the authors’ method for confirming their results?

- The discussion section is poorly written. The authors have used only four references in this section! More references are needed to support and strengthen the statements/conclusions/predictions in the discussion (Interpretation of the results by the authors, and comparison of the results with the available evidence are required.)

Minor Comments:

- Under “Discussion” establish a new heading “Implication of the study” and explain the theoretical contribution and practical implication of this study.

- The conclusion section must be moved after the discussion and implication of the study (discussion, implication of study, conclusion!)

- Some of the Tables can be moved to the supplementary information section (check the author's instructions for details).

7. PLOS authors have the option to publish the peer review history of their article (what does this mean?). If published, this will include your full peer review and any attached files.

Reviewer #1: No

Reviewer #4: No

---

## [Author Response · Author response to Decision Letter 1]

3 Mar 2023

Dear Reviewers,

Thank you very much for your comments and suggestions on the paper "Mining and quantitative evaluation of COVID-19 policy tools in China". I will reply to each reviewer's comments one by one.

Many thanks to reviewer 1 for pointing out detailed errors in format of tables. I have modified the tables. In addition, I added the methodology section and introduced the expression PMC-AE. With your comments in mind, the summary has also been revised.

Many thanks to reviewer 4 for your valuable suggestions. The manuscript has been refined for English grammatical structure and phraseology. The abstract has been reworked to highlight the main findings and conclusion. According to your suggestions, I have readjusted the structure of the article and clearly explained the data collection process, methodology and results, which makes the article more clearly expressed. The discussion section adds more references and also contains some comparisons of the results with the available evidence, which includes how China has used a mix of policy instruments, how this mix has changed over the course of the crisis, and to compare it with previous studies. Because the theoretical contributions and practical implications of the research have already been mentioned in the introduction section, they are not mentioned again in the discussion section. The conclusion section is moved after the discussion of the study.

Finally, I would like to express my sincere thanks again. I hope that you can put forward valuable comments on my revised article again and look forward to your reply.

Sincerely,

Luming Cheng

School of Economics and Management, Tianjin Chengjian University, Tianjin, China

chengluming@yeah.net

---

## [Decision Letter · Decision Letter 2]

27 Mar 2023

Mining and quantitative evaluation of COVID-19 policy tools in China

PONE-D-22-20862R2

Dear Dr. Cheng,

We’re pleased to inform you that your manuscript has been judged scientifically suitable for publication and will be formally accepted for publication once it meets all outstanding technical requirements.

Kind regards,

Peyman Rezaei-Hachesu, Associate professor

Academic Editor

PLOS ONE

Additional Editor Comments (optional):

Reviewers' comments:

Reviewer's Responses to Questions

**Comments to the Author**

1. If the authors have adequately addressed your comments raised in a previous round of review and you feel that this manuscript is now acceptable for publication, you may indicate that here to bypass the “Comments to the Author” section, enter your conflict of interest statement in the “Confidential to Editor” section, and submit your "Accept" recommendation.

Reviewer #4: All comments have been addressed

2. Is the manuscript technically sound, and do the data support the conclusions?

Reviewer #4: Yes

3. Has the statistical analysis been performed appropriately and rigorously? 

Reviewer #4: Yes

4. Have the authors made all data underlying the findings in their manuscript fully available?

Reviewer #4: Yes

5. Is the manuscript presented in an intelligible fashion and written in standard English?

Reviewer #4: Yes

6. Review Comments to the Author

Reviewer #4: (No Response)

7. PLOS authors have the option to publish the peer review history of their article (what does this mean?). If published, this will include your full peer review and any attached files.

Reviewer #4: No

---

## [Editor Report · Acceptance letter]

29 Mar 2023

PONE-D-22-20862R2 

Mining and quantitative evaluation of COVID-19 policy tools in China 

Dear Dr. Cheng:

I'm pleased to inform you that your manuscript has been deemed suitable for publication in PLOS ONE. Congratulations! Your manuscript is now with our production department. 

Kind regards, 

on behalf of

Dr. Peyman Rezaei-Hachesu 

Academic Editor

PLOS ONE